# Edible Film Casting Techniques and Materials and Their Utilization for Meat-Based Product Packaging

**DOI:** 10.3390/polym15132800

**Published:** 2023-06-24

**Authors:** Fauzi Atsani Harits Prakoso, Rossi Indiarto, Gemilang Lara Utama

**Affiliations:** 1Faculty of Agro-Industrial Technology, Universitas Padjadjaran, Jalan Raya Bandung Sumedang Km. 21 Jatinangor, Sumedang 45363, Indonesia; fauzi15001@mail.unpad.ac.id (F.A.H.P.); rossi.indiarto@unpad.ac.id (R.I.); 2Centre for Environment and Sustainability Science, Universitas Padjadjaran, Jalan Sekeloa Selatan I No. 1, Bandung 40134, Indonesia

**Keywords:** biopolymer, meat products, edible film, shelf life

## Abstract

According to a profusion of academic studies on the use of organic materials or biopolymers as key components, the current trajectory of food packaging techniques is showing a positive inclination. Notably, one such biopolymer that has attracted much attention is edible film. The biopolymers that have been stated as constitutive components are composed of polysaccharides, lipids, proteins, or a combination of these, which work together to reinforce one another’s properties and create homogenous mixtures. An edible film provides a clear, thin layer that encases foodstuffs, including their packaging. The production and use of edible film have recently been the focus of much research in the field of food polymers. Extending the shelf life of food goods is the goal of this research. Given their great susceptibility to change brought on by outside forces or pollutants, which may result in oxidative rancidity, the proper storage of nutrient-dense food items, particularly meat products, deserves careful study. Many edible films have been found to contain active ingredients, such antimicrobials or antioxidants, that can successfully prevent the spoiling of meat products, a process that can happen in a short amount of time. Surprisingly, a number of scholarly examinations reveal that edible film may be cooked alongside meat because of its organic makeup. We hope that the use of edible film will lead to a more environmentally responsible method of food packaging than has previously been possible.

## 1. Introduction 

The consideration of meat product packaging is of utmost importance since it plays a crucial part in establishing whether an edible item is suitable for consumption. Meat products’ exposure to the environment directly contributes to their shorter shelf life. The primary goal of food packaging is to maintain the product’s quality and safety, guaranteeing that it may be consumed without posing any health risks [1]. The use of packing materials in the preservation of meat products is intended to prevent the growth of bacteria, obstruct oxidative processes, mask offensive qualities, and maintain the food’s nutritional integrity [2,3].

Throughout the whole production process until they are prepared for public consumption, meat products meant for consumption must be kept in the best possible condition. It is insurmountably difficult for suppliers of foods to market products that experience such degradation given the intrinsic constraints of food spoilage [3]. It is crucial that packaging not only complies with the requirements for extending the product’s shelf life, but also demonstrates the capacity to protect its contents from the point of manufacture to storage, sale, and consumption by the general public. The convenience of the food’s packing is what makes it unique [4].

At the moment, most food packaging is made of manufactured plastics that are not very good for eating. Additionally, it is important to keep in mind that plastic packaging is very bad for the Earth because it is hard to break down and can take hundreds of years [5]. In recent years, there has been a rising trend toward using edible films as a way to package meat [6,7]. The main goal of making edible films is not to completely replace synthetic plastics, but to offer a more eco-friendly option for food packaging [8]. To make edible film, a food-safe solution is used to make a thin layer. There are two ways to make a tasty film for this purpose. The first method focuses on drying at a certain temperature, which causes water in a suspension to evaporate and form a thin layer. The second method focuses on applying heat and pressure to a raw material at the same time, which forces it out of the extruder and forms a thin layer [9,10]. At the moment, food items are coated on the outside with edible film [11]. Biopolymers, such as polysaccharides, lipids, and proteins, are used to make edible films because they are easily biodegradable, plentiful, and sustainable in nature [12].

Without the right packaging, high-nutrient meat products can be ruined by the growth of microorganisms and reactions with oxygen in the air. The term “deterioration” refers to the harmful changes that can take place in meat goods [13]. The use of glucose by microorganisms to make energy and their following growth into colonies causes meat products to go bad [13,14]. When meat products oxidize, their fats and proteins change, and browning reactions and microbial growth happen. These changes have a considerable effect on the way the meat tastes and how it feels [15,16]. Given that meat goods have higher economic value than most other food items, it is of utmost importance to ensure that they maintain high quality [17].

The packaging used for meat preservation is responsible for protecting it against physical, chemical, and microbial contamination [15]. Due to the reduction in microbial activity, it has been found that maintaining a temperature range of 2–5 °C, along with appropriate packaging, can increase the shelf life of beef products [18]. In addition, combining the use of the two methods can result in an improved method of packing and preservation [19]. The main goal of this article is to investigate the manufacturing of edible films, the biopolymers used in their casting, how these films are used in meat products, and how these factors affect the longevity of meat products.

## 2. Edible Film Casting Techniques

### 2.1. Wet Formation (Solvent Casting)

One approach to casting an edible film is through wet formation or solvent casting. The basic idea behind this technique is to evaporate water-containing biopolymers to dry the solution [20]. The technique in question is the most widely used method for creating edible films (Figure 1). Four key phases are commonly included in the process for creating edible films: dissolving biopolymers in solvents, applying heat treatment, casting, and drying [21,22]. The biopolymer and solvent are combined in the first step, and then, the chemical links between them are broken through heat treatment to help the biopolymer dissolve in the solvent [23]. The amalgam is then homogenized and cast in an appropriate matrix that can produce an edible film with the desired shape. It is common practice to measure the volume of the solution to determine the final edible film’s thickness. The solvent must then be allowed to evaporate in order to desiccate the film. In order to avoid excessive desiccation and structural damage to the edible layer, the process of evaporation often takes place inside the confines of the oven, where temperatures are carefully controlled to keep them below 60 °C. The film can be delaminated and used for its intended purpose when the solution has evaporated and dried [10].

It has been noted that using a short time frame and a high temperature during the drying phase of wet formation did not produce a film with superior mechanical or structural characteristics [24]. Therefore, it is crucial to remember that the drying process should take longer than 12 to 24 h to ensure the best possible evaporation of the solution. Improved functional performance may result from the use of edible coatings with enhanced mechanical and structural properties [21].

Wet casting is a better technique for creating edible films made of polysaccharides because of the material’s hydrophilic qualities. It has been suggested that shielding the edible film from harm during the peeling procedure may make it easier to remove the film from the mold [25]. The polysaccharide in question, which includes both amylose and amylopectin, has a variety of advantageous characteristics in the context of producing edible films. These two substances serve as binders and viscosity enhancers after being heated in liquid form with other ingredients [26]. In order to create edible films, starch is heated while being mixed with a solvent, such as water or alcohol. Amylose and amylopectin both gelatinize at a specific temperature. Enhancing the solubility of polysaccharides during manufacture is the goal of the gelatinization process. By releasing amylose from the polysaccharide granules and spreading it in the solvent, this can be accomplished [22].

Figure 1 shows the typical procedure used to produce edible films. The solubility of biopolymers in solvents, the application of heat treatment, the printing process, and the drying process are typically the four basic phases in the formation of edible films [21,22]. The biopolymer and solvent are combined in the first step, and then, the chemical links between them are broken via heat treatment, allowing the biopolymer to dissolve more easily in the solvent [23]. After being homogenized, the amalgam is cast in an appropriate matrix that may create an edible film with the required geometry. To calculate the thickness of the resulting edible film, it is common to measure the volume of the solution. The solvent needs to evaporate for the film to finish drying. In order to avoid severe desiccation and structural damage to the edible film, evaporation often takes place within the oven’s confines, where temperatures are carefully managed to keep them below 60 °C. The film can be delaminated and used for its intended purpose after solution evaporation and drying [10].

Using a short time frame and a high temperature during the drying phase of wet formation did not produce a film with superior mechanical or structural characteristics [24]. Therefore, it is crucial to remember that the drying process should take longer than 12 to 24 h to ensure the best possible evaporation of the solution. Improved functional performance may result from the use of edible coatings with enhanced mechanical and structural properties [21].

### 2.2. Dry Formation Casting Techniques

The dry formation method requires the presence of minute quantities, if any, of substances other than the primary constituent because there is no liquid component [27]. The basic idea behind this process is illustrated in Figure 2, where heat energy is applied to the powdered substrate to create a polymeric film [9]. The powder is first added to the feed hopper to begin the dry forming process. The biopolymer powder is then compressed using a revolving screw, which aids in the creation of a homogeneous solid material unit and functions similarly to stirring. Incorporating a plasticizer during this step is typical to speed up the compaction process. The heat treatment begins and lasts for the specified amount of time at temperatures between 75 and 100 °C [27,28].

Plasticizers are frequently required when using the dry method for edible film casting to increase the films’ flexibility and durability. Plasticizers are applied to polysaccharide- and protein-based films and coatings to reduce their natural brittleness [29]. The mechanical characteristics of the film are improved by plasticizers, while the increase in film permeability is kept to a minimum [30]. The following types of plasticizer are frequently used in the production of edible films: (1) monosaccharide-based plasticizers, such as glucose and fructose; (2) oligosaccharide-based plasticizers, such as maltodextrin; (3) polyol-based plasticizers, such as glycerol and sorbitol; (4) lipid-based plasticizers, such as vegetable oil; and (5) various derivatives of monosaccharide. Starch films can have plasticizers added to them to increase their flexibility and decrease their brittleness [31]. The film’s barrier qualities can be compromised by the addition of plasticizers, which can also make the film more permeable [32]. 

The mixture of powdered components and plasticizer will transition into a molten state inside the defined “molten plastic” region as a result of the heat energy and mechanical agitation provided by the revolving screw. This molten condition will help to speed up the edible film casting process. Due to its link with the shear rate and shear stress during the extrusion process, the rotating velocity of the screw has a considerable impact on the resulting film [28,31]. The mechanical characteristics of polysaccharide films have been found to suffer as the screw’s rotation speed is increased. It is crucial to remember that the rotating screw’s speed needs to be adjusted depending on the particular material being used [33].

The layered film configuration is favored for the current technique, which aims to generate a film fit for consumption. The film in question cannot be approved for usage after the formation process. The film must go through additional processing, specifically, conditioning, at specific temperature and humidity levels, such as conditioning comprising heating the produced film to 60 °C for two hours in an oven. A different method of conditioning the produced film, however, involves exposing the extruded film to a relative humidity of 56.7% for a period of 7 days [31].

This technique’s accelerated processing time compared to the wet formation procedure is one of its significant advantages. Wet formation requires a 12 h drying process at room temperature for the biopolymer and solvent to be homogenized [34]. Additionally, this method has the ability to produce a bigger number of edible films at a lower cost, making it a workable method for commercializing edible films [35].

## 3. Biopolymer Materials for Edible Films and Their Application to Meat Products

### 3.1. Polysaccharides

Numerous polysaccharide biopolymers come from different plant sources [36]. Due to their many advantageous qualities for food packaging, polysaccharides have gained a lot of popularity as a preferred option for edible film creation (Table 1). These include the capacity to produce a film that is more transparent than other biopolymers and is odorless, preventing the sensory perception of food from being hampered [37]. Polysaccharides undergo edible transformations that give them commendable oxygen-holding qualities because they include several hydrogen chains that add density and prevent gas ingress and egress [38]. However, the polysaccharide’s hydrophilic nature gives it remarkable resistance to watery settings [39]. Polysaccharides are a notable class of compound and are frequently used as basic constituents for the creation of edible films. Figure 3 depicts the chemical composition of polysaccharides, which are frequently utilized as ingredients in edible film. The components mentioned include cellulose, chitosan, starch, and gum arabic [37].

#### 3.1.1. Starch

Being a readily available polysaccharide, starch is a very affordable alternative. It is interesting that the starch content of wastes and byproducts from the food sector can be used [40]. Starch is used to create films, and these films have unique characteristics such as increased visual transparency and the absence of any detectable gustatory or olfactory sensations [41,42]. Despite the benefits indicated above, it is commonly accepted that starch-based films have poor water- and moisture-barrier qualities because of their natural hydrophilicity. Therefore, using plasticizers and other polymers to make up for the shortcomings in their intrinsic characteristics is a typical practice [43].

**Figure 3 polymers-15-02800-f003:**
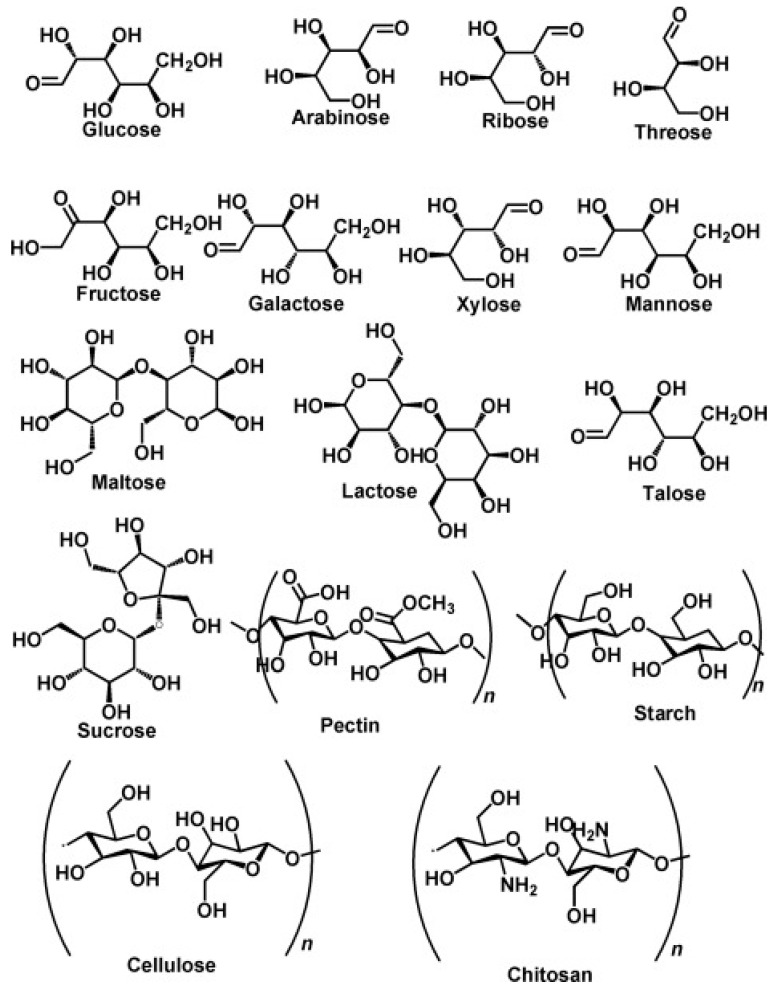
The chemical structure of polysaccharides that are commonly used as edible film materials [44].

Plasticizers must be carefully considered because it is crucial to ensure that the edible film preserves its palatability and that its consumption does not endanger humans. Given that starch-based edible films with high starch concentrations exhibit considerable brittleness, plasticizers are added to them in an effort to increase their flexibility and pliability [45]. Starch is a common byproduct of the film creation process, which involves wet formation and heat treatment. When starch is heated, it goes through a swelling phase that occurs between 60 and 80 °C, because the starch granules begin softening at this point [46]. Given these facts, it is interesting that the plasticizer has the ability to penetrate and intercalate inside hydrogen chains, causing a relaxation effect that, in turn, grants the film greater elasticity [47]. Due to stronger hydrogen bonds, a starch that contains more amylose than amylopectin generates a viscous gel and typically needs a higher temperature. Retrogradation is the process of reassembling higher-amylose-content starch after it has been gelatinized due to its linear molecular structure [48].

**Table 1 polymers-15-02800-t001:** Edible films derived from polysaccharides and their application to meat products.

Starch Type	Other Materials	Meat Product	Effects	References
Corn starch	Gelatin	Chicken breast fillet	Control microbial growth	[49]
Corn starch	Essential oil	Red meat	Antimicrobial and antioxidant effects	[50]
Cassava starch	Essential oil	Ground beef	Antimicrobial and antioxidant effects	[51]
Potato starch	Essential oil	Pork	Antimicrobial and antioxidant effects	[52]
Potato starch	Sea buckthorn	Beef	Reduce oxidation	[53]
Cellulose	Essential oil	Beef	Antimicrobial, control pH value	[54]
Cellulose	Probiotic	Chicken	Antimicrobial	[55]
Cellulose	Alginate	Chicken	Reduce oxidation	[56]
Chitosan	Essential oil	Chicken	Reduce microbial activities	[57]
Chitosan	Casein	Chicken	Control oxidation	[19]
Gum	Soy protein	Chicken breast	Control oxidation	[13]

The simplicity with which components derived from starch dissolve in glycerol indicates the affinity of glycerol for starch [58]. Glycerol is added to edible film to increase its viscoelasticity; however, excessive glycerol use is not advised. Glycerol concentrations beyond 50 percent have the potential to weaken polymer bonds, decreasing the permeability of edible films [59,60]. In addition to glycerol, there are other plasticizers that are used in the production of edible films. These include: (1) sorbitol, which is sugar alcohol that can enhance the flexibility and mechanical properties of the film; (2) certain monosaccharides and oligosaccharides, including sugars like glucose, fructose, and maltose; (3) polyols, such as mannitol and xylitol, that contribute to the film’s flexibility and water vapor barrier properties; (4) hydroxypropyl methylcellulose (HPMC) and carboxymethyl cellulose (CMC), which are cellulose derivatives that can enhance the film’s mechanical and water vapor barrier properties; and (5) lipids, including waxes, oils, and derivatives, which improve the film’s flexibility and water resistance [61,62,63,64].

Along with plasticizers, combining starch with protein or lipids to create a homogenous mixture of edible film can have further benefits [42]. The incorporation of these components aims to improve both the functional qualities and the mechanical qualities of films. Utilizing essential oils has been found to promote antibacterial properties, extending the shelf life and guaranteeing the sterility of meat products [52]. Corn starch-based edible films strengthened with 3% essential oil have the capacity to prevent the proliferation of *Pseudomonas sp.*, while the addition of gelatin to a starch-based edible film for chicken fillets results in chicken fillet with gelatin’s intrinsic freshness, which successfully prevents any appreciable change in pH levels [50]. It is a known truth that gelatin has the power to prevent lipid oxidation, a process that raises pH levels and serves as a warning sign that meat that has lost its freshness [65]. Protein and polysaccharide amalgamation’s hermetic nature causes a compact homogenous mixture structure to form, which promotes the growth of lactic acid bacteria [49].

#### 3.1.2. Cellulose

Cellulose-based materials are abundant and come from a range of sources, such as leaves, food waste, and microorganisms [66]. Due to its distinct crystalline structure that distinguishes it from the bulk of polysaccharides, the substance known as cellulose provides a difficult challenge in terms of solubility in aqueous solutions. Recent developments in the field of cellulose derivatives have produced a number of goods with improved water solubility: methyl cellulose (MC), carboxy methyl cellulose (CMC), hydroxyl propyl cellulose (HPC), and hydroxyl propyl methyl cellulose (HPMC) [36]. The derivative products display traits that are analogous to those of polysaccharides in general, such as transparency, odorlessness, and tastelessness [67].

The use of polysaccharide-based films in meat products requires the introduction of additional ingredients to improve their functional qualities. In this instance, spice extracts were employed due to their phenolic compound content, such as flavonoids, which possess remarkable antimicrobial and antioxidant characteristics that can contribute to the functional attributes of edible films [68]. Meanwhile, adding rosemary extract to cellulose-based film can successfully extend the freshness of red meat for a period of 12 days, because the flavonoids in rosemary have been found to prevent bacteria from synthesizing nucleic acids in their cytoplasm, which disrupts their metabolic processes and eventually causes them to stop growing [54,69]. The antioxidant activities of flavonoids aid in the preservation of meat’s freshness. Because of the way flavonoids work, the lipidic fraction of beef liposomes is protected from oxidants’ damaging effects [54].

#### 3.1.3. Chitosan

Chitosan is a polysaccharide of animal origin sourced from marine organisms, particularly shrimp. Chitosan, like other polysaccharides, exhibits film characteristics that are characterized by brittleness [70]. Additionally, chitosan displays impressive antibacterial and antioxidant properties after the deacetylation process, making it a highly effective packaging material. When chitosan is deacetylated, chitin derivatives derived from shrimp are produced that contain the antibacterial and antioxidant compounds β-(1-4)-2-acetamido-D-glucose and β-(1-4)-2-amino-D-glucose. Fresh chicken meat’s shelf life in refrigerators has been found to be greatly extended for a period of 12 days by using edible film made of chitosan [35]. Chitosan edible film has also shown inhibitory effects on the growth of *L. monocytogenes* and *E. coli* on the surface of meat [71]. Furthermore, using chitosan in the production of edible films at concentrations between 0.5 and 1.5% results in meat products that exhibit no malodorous attributes even after a lengthy 9-day storage period at 4 to 7 °C [72].

#### 3.1.4. Gum

Gum is frequently employed in a range of applications as a thickening agent, emulsifier, or stabilizer [73]. Acacia gum is the gum that is most commonly used in commercial applications among the wide variety of gums [3]. The revered *Acacia niloticalinn* contains exceptional antibacterial properties, most notably the ability to prevent the spread of *Aspergillus niger*, *Candida albicans*, *Micrococus luteus*, and numerous other microorganisms [74].

Gum’s use in meat products requires its interaction with other substances, such as essential oils, in order to properly activate its antioxidant properties and strengthen its antibacterial properties. Edible gum-based films infused with essential oils from cinnamon and garlic have antimicrobial qualities that can successfully prevent microbial activity in fish that has been preserved for up to 18 h [73]. Additionally, gum has been found to have the ability to prevent the growth of bacteria in chicken filets preserved at a temperature of 4 °C for a period of 21 days without the addition of any other compounds to the film base material. When chicken flesh was coated with a gum-based film at a concentration of 25%, it showed a microbial count of 0.8 × 10^2^ cfu/g compared to the untreated chicken meat’s 25 × 10^6^ cfu/g [74]. 

### 3.2. Lipid

Lipids, as a secondary biopolymer, are commonly employed as a constituent in edible films. The primary origins of lipids are animal and vegetable fats [3]. It is a widely accepted practice to employ vegetable oils as edible films additives [70,75]. The hydrophobic nature of edible lipid film, owing to its lengthy fatty acid chains, renders it an excellent water vapor barrier. As a result, it is a highly desirable component in meat products, as it effectively mitigates fat oxidation [3,76]. Nevertheless, lipid utilization results in the formation of a film that is typically thicker than conventional ones. In addition to their high viscosity, the resultant films often exhibit brittleness, thereby mandating the incorporation of biopolymers or other adjuncts to enhance their mechanical characteristics. 

#### 3.2.1. Wax

One example of a substance generated from lipids is wax. From a commercial standpoint, wax is frequently produced using petroleum. It is important to remember that biotic sources, such as plants and wildlife, can provide wax [77]. It is generally agreed upon that using waxes derived from biological sources is a safer and more preferable method for creating edible films. Beeswax and plant-based carnauba wax are the two most common types of wax; however, there are many others, as well [4,78].

Beeswax is used in edible films to give them a distinctive yellow tint because it comes from the *Apis mellifera* species’ abdominal glands, which are also used to make honeycomb. In the context of its practical use, beeswax exhibits greater ease of removal due to its comparably reduced adhesive force to other lipidic compounds [79]. A thick layer may need to be removed before ingestion when using wax as an edible film. This is owing to the worry that its removal might conceivably change the food’s flavor character. It is considered permissible to persist and ingest the resulting film layer if it is thin [77].

#### 3.2.2. Vegetables Oils

In the production of edible films, vegetable oil serves several roles, such as plasticization, preventing gas permeation, facilitating adhesion to the food surface and contributing water resistance, and providing flexibility. Vegetable oil could increase a films’ flexibility, stretchability, and overall mechanical properties, and lessen the fragility of the film and increase its processability during casting [80]. Due to its liquid nature and high lipid content, vegetable oil can effectively plasticize the matrix of a film [75]. The incorporation of vegetable oil into the production of films with enhanced gas and moisture barrier properties is possible. It aids in decreasing the film’s gas and water vapor permeability, thereby preserving the quality and freshness of food products [81]. Vegetable oil aids in the even distribution of the film-forming components, thereby facilitating the formation of a uniform and seamless film structure. It facilitates adhesion between the film and the food surface and ensures adequate coverage, and also acts as barrier that can prevent the transfer of volatile compounds from the film to the food [82].

However, food products with oxidized lipids generate rancid off-flavors, making them unfit for human ingestion [19]. Rancidity produces harmful aldehydes and degrades polyunsaturated fatty acids, reducing nutritional value [83]. Food packaging with edible films and coatings protects and preserves food. However, rancid oil additives in these films can alter the taste and quality of packaged food. Rancid oils can also cause food spoilage. Meanwhile, essential oils in edible films may improve food shelf life, and their antibacterial characteristics include fighting pathogens and rotting organisms [84]. However, the addition of vegetables oils to edible films mostly degrades foods’ sensory quality, nutritional value, and shelf life, which is why vegetables oils are no longer used as additives for edible film casting.

### 3.3. Protein

In the creation of edible films, protein, the third biopolymer, is used (Table 2). In contrast to polysaccharides and lipids, protein has been shown to have outstanding mechanical characteristics [85]. Given its advantageous hydrophilic characteristics, the proteinaceous edible film in question makes an excellent surface for products that need to be wrapped or coated with edible film. Furthermore, it has remarkable O_2_ and CO_2_ barrier qualities. It regretfully does not have any water diffusion resistance. Proteins’ water resistance properties can be efficiently changed by adding hydrophobic materials like beeswax or plasticizers like glycerol [86,87]. 

Proteinaceous edible films have been found to have the potential to form intermolecular links in a variety of orientations, which results in the creation of a stiffer edible film architecture [86]. For the creation of edible films generated from animals, protein sources such whey protein, casein, gelatin, and egg albumin have been found to be promising candidates. A number of plant-based sources, including, but not limited to, peanuts, soybeans, corn, rice, and wheat, have been mentioned as being suitable for use in food polymer applications [86,88].

**Table 2 polymers-15-02800-t002:** Edible films derived from protein and their application to meat products.

Protein Type	Other Materials	Meat Product	Effects	References
Whey protein isolate	Essential oil	Sausage	Antimicrobial, control sensory quality	[89]
Whey	Seaweed extract	Chicken	Antioxidant, reduce lipid oxidation	[90]
Whey	Palm oil	Chicken	Antioxidant, reduce lipid oxidation	[91]
Gelatin	Corn starch	Chicken breast fillet	Control microbial growth	[49]
Gelatin	Henna extract	Beef	Antioxidant, antimicrobial, control sensory quality	[92]
Gelatin	Transglutaminase enzyme	Beef	Antioxidant, antimicrobial, control sensory quality	[93]
Soy protein	Carboxymethyl cellulose	Pork	Antioxidant, control moisture loss	[94]

#### 3.3.1. Whey Protein

Whey is a leftover protein-rich byproduct of cheese-making processes that use milk as their main substrate. Whey protein is usually referred to as a milk protein when combined with casein. Films made from whey display remarkable gas barrier qualities and have the capacity to hold onto aroma. However, because polysaccharides are hydrophilic, it is essential to add more hydrophobic components to improve their water and vapor resistance qualities [95]. Whey protein, a milk-derived product, is known to contain the proteins immunoglobulin, lactoglobulin, and lactalbumin, which have been shown to give packaging film antibacterial capabilities, making it an active packaging material [37].

The use of whey protein that has been fermented with *Candida tropicalis* to produce edible films that exhibit superior functional qualities in the presence of bioactive peptides produced from microorganism degradation is a recent advancement in the field of active antimicrobial packaging [96]. These microorganisms have the capacity to catalyze the breakdown of proteins, which results in the synthesis of strong antimicrobial substances such lysozyme, lactoferrin, and lactoperoxidase [81]. *Candida tropicalis* breaks down the lactoferrin peptides in the fermented cheese whey to produce lactoferricin, which has strong antibacterial effects [97,98]. It is known that the presence of lactoferrin prevents microbial growth by adhering to the cell surface, rupturing cell membranes, and hindering the function of cytoplasm in germs.

Meanwhile, applying edible films made of whey protein to salmon during cold storage resulted in a significantly lower loss of moisture than in a control group [6]. The use of whey-based edible films with a 13% whey protein concentration in their formulation has demonstrated a notable decrease in lipid oxidation in kilka fish [99]. On the other hand, essential oils have been found to improve the antibacterial effectiveness of whey protein, especially towards *Pseudomonas sp.* in chicken meat, which has been found to be significantly suppressed [6].

#### 3.3.2. Gelatin

Gelatin is a frequently used substrate for edible films because of its easy availability and great mechanical qualities. Due to its extraordinary ability to gel, it may be processed both wet and dry, which increases its pliability during the development of an edible film [95,100]. Collagen from the dermal tissues of different animal species, including bovine, porcine, and aquatic species, is usually credited with being the source of gelatin. Fish gelatin is seen as being a safer option than bovine gelatin due to the potential presence of parasites in some cattle skin, creating worries about its safety. Pork gelatin is a common source of gelatin; however, it is incompatible with Muslim consumers, so it is wise to use caution when using it widely given Islamic dietary constraints [35].

The application of gelatin coating to beef patty samples, compared to a control treatment, resulted in the lowest degree of shrinkage, according to the results of a 30-day refrigerator storage test [100]. The hydrophilic amino acids found in gelatin are thought to be the cause of this phenomenon since they have a remarkable capacity to bind water, which prevents meat from shrinking [92]. Furthermore, gelatin-based films have been shown to have the ability to limit microbial growth, as seen by the preservation of a pH level that is reasonably stable in pig products, compared to films made of traditional plastic materials [101].

#### 3.3.3. Soy Protein

Soy protein is a byproduct of the refining of soy oil, much like whey protein is. Soy protein has gained a lot of traction as a plant-based substitute for meat because of its texture, which is similar to that of meat [66]. Approximately 90% of the soy protein utilized in the production of edible film building ingredients is soy protein isolate, which undergoes processing to increase its protein content. Because of its large molecular weight, soy protein has admirable film properties that lead to outstanding tensile strength and elongation break [86]. In comparison to other protein types, soy protein has a higher degree of translucency in its chromaticity, and its texture has a more velvety feeling. Because soy protein has poor thermal stability and dry formation techniques need greater temperatures than moist formation, these techniques are not suited to soy protein [3].

On the other hand, the utilization of edible films made of soy protein has the potential to reduce sausage weight loss. Edible films made of protein have denser characteristics, which makes it easier to properly preserve evaporated water during storage [102]. The protective action of a coating made of soy protein is responsible for the preservation of hog meat’s attractive red color. In situations where meat is judged to be in ideal condition, a sizable proportion of myoglobin pigment persists, giving the meat its distinctive red color. One well-known effect of lipid oxidation is the gradual loss of myoglobin pigment in meat, which results in its coloring [94].

## 4. Advantages and Disadvantages

It is crucial to consider certain qualities of edible films in relation to their functional capabilities when discussing how to increase the shelf life of meat products. This is especially important given how quickly oxidation and degradation start to occur in such compounds. The key characteristics that support edible films’ effectiveness in extending meat’s shelf life include their permeability to oxygen and water vapor, and their antibacterial and antioxidant capabilities [21,103]. Water vapor permeability (WVP) is a crucial property that acts as an indicator of meat condition. It is interesting to note that a high WVP is linked to the film having a lot of pores, which raises the possibility of gas exchange. In turn, this exchange interacts with lipids and starts the oxidation process [104]. Even though edible film has no pores and completely covers the surface of the meat, the inevitable oxidation still poses a problem. Antioxidants’ importance in this setting simply cannot be emphasized enough [38,105]. On the other hand, it is recognized that using antimicrobials can prevent meat deterioration brought on by microorganisms [11].

To achieve desirable water vapor permeability properties in edible films, it is important to reduce pore formation in edible films. The key to solving the problem is to combine the right elements using casting techniques. Due to their hydrophobic properties, lipid-based materials, such waxes, have been used as additives to reduce the permeability of water vapor [105]. The production of edible films made from whey and wax amalgams, carried out using the arid approach and an extrusion device, is a notable example of effective amalgamation to reduce WVP levels [106]. The concentration of wax has a considerable effect on the quality of these edible films, according to an investigation of the edible film’s microstructure. In particular, a smoother and denser film surface is produced by a higher wax concentration. The best way to create films with reduced water vapor permeability is to combine the desiccation casting technique with the addition of lipid adjuncts to the film matrix. During the extrusion process, lipids are exposed to high pressure and temperature, which causes structural degradation and the subsequent amalgamation of those lipids with other elements, culminating in the production of a highly compacted matrix. Lipids are less able to interact with steam or water because of their hydrophobic nature [107]. Lipids’ ability to maintain the consistency and flavor of meat is one of the key benefits of using them in the composition of edible films for meat products. The flavor characteristic of processed beef products covered with films that contain wax frequently changes during cooking [108], unlike films made of starch and protein that lack lipids in their composition. Although the extrusion process is still used to create edible films, it is important to note that due to the hydrophilic nature of the materials used, the finished product may not have the desired water vapor permeability [86,87]. The extrusion process to create edible films based on starch led to undesirable water vapor permeability (WVP) qualities [31]. Water vapor permeability (WVP) was insufficient as a result of using sodium caseinate in the extrusion method to create edible films [109]. The hydrophilic properties of casein, which was used as a main material, are responsible for this result. Additionally, the incorporation of glycerol, a plasticizer with hygroscopic properties, made it easier for the films to absorb water vapor. Hydrophilic protein-based films may not completely prevent the shrinking of both fresh and processed meat when using edible film for meat product packaging [110]. However, whey protein has a positive quality, namely, its ability to act as an antimicrobial agent. It is well known that milk protein is made up of a variety of biologically active substances, such as α-lactalbumin, β-lactoglobulin, immunoglobulins, bovine serum albumin, and protease peptones, which have been shown to have strong antimicrobial effects on a variety of microorganisms [3]. Furthermore, whey protein is found to possess lysozyme, which is an efficacious antimicrobial agent with the potential to mitigate microbial contamination [111].

The hydrophilicity of the constituent materials is crucial since the basic tenet of the wet method calls for the dissolution of the constituent elements in the solvent. It is crucial to mention that using lipids in this particular process is not thought to be appropriate. Either the dry approach or the extrusion method will go into greater depth on the lipid component. The primary proteinaceous substrate used in collagen’s incorporation into meat products is collagen [112]. Collagen performs sub-optimally in sustaining gaseous exchange, as is frequently seen in proteinaceous films. Nevertheless, there are several advantages connected to collagen’s interaction with meat products, especially in the context of foods made from chicken. It is conceivable to apply collagen directly to the surface of fresh meat in order to preserve the raw state of the animal. Collagen has been shown to be an excellent inhibitor of microbial growth. Additionally, it has been noted that poultry meat has the ability to stop lipids from seeping into its tissue while cooking [113].

In addition to protein, polysaccharide-based ingredients, such as chitosan, can be used to extend the time that meat can be stored. Chitosan is a versatile material that may be made using either dry or wet methods [3]. It has been discovered that using wet-method chitosan as a coating agent for pig patties effectively extends the shelf life of the product because of chitosan’s natural antibacterial capabilities. However, the addition of sunflower oil to the system lowers its antimicrobial effectiveness [70]. 

Cellulose also presents itself as an alluring polysaccharide substrate, in addition to chitosan. In the world of food polymers, carboxymethyl cellulose, sometimes known as CMC, is a prevalent type of cellulose. Electron microscopy analysis of the wet approach for the fabrication of CMC-based films revealed a noticeably coarse surface [114]. The surface roughness observed can be attributed to the relatively substantial particle dimensions of carboxymethyl cellulose (CMC), coupled with the inherent recalcitrance of cellulose towards aqueous dissolution. Due to its weakly connected polymers, the resulting film is thought to be less effective at preserving chicken sausages within a refrigerator, affecting the sausage’s ability to retain moisture. The influence of microorganism activity also affects the pH of sausages, making them susceptible to fast change. Providing supplemental ingredients like whey protein or essential oils is important in specific situations. Meanwhile, CMC-based films made using the wet approach revealed a noticeably uneven surface [115]. The unusually large particle sizes of carboxymethyl cellulose and cellulose’s innate characteristics, which prevent it from dissolving in water, together account for the observed surface roughness. Due to the poor intermolecular interactions of the polymers, the resulting film is regarded as inadequate for protecting chicken sausages in a refrigerator and maintaining their moisture content. The microbial activity causes the pH of the sausage to change quickly. To avoid the aforementioned occurrence, it may be required to include supplemental ingredients in some situations, such as whey protein or essential oil [57,116].

## 5. Conclusions

The goal of edible films is to provide a more environmentally sound substitute for the non-green polymers that have previously dominated food packaging. It is important to highlight that the use of edible films not only extends the shelf life of food goods but also enhances a wide variety of consumables packaging, not just meat products. Either wet or dry formation can be used to create edible films; notably, the wet forming approach lends itself better to the creation of water-soluble biopolymers. However, it is important to note that although it requires a significant time commitment, the dry forming method produces better results in terms of mechanical qualities. It is conceivable to use biopolymers, such as polysaccharides, lipids, and proteins, in the casting of environmentally friendly packaging for foods, especially meat products. It is crucial to remember that each of these biopolymers has a unique set of limitations. To overcome these limitations, additional materials like co-biopolymers, plasticizers, essential oils, and other compounds of a similar nature must be used. It is interesting that adding edible coatings can greatly increase the shelf life of meat products. This is because edible films have a variety of uses, including moisture retention, water binding, microbial activity control, controlling oxidation, and more. The food sector has a strong desire to start using edible food packaging given the vast number of studies that have been conducted on edible films.

## Figures and Tables

**Figure 1 polymers-15-02800-f001:**
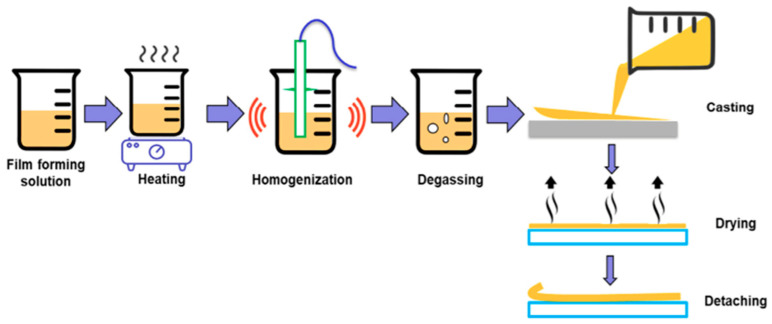
Wet casting techniques for edible film formation [10].

**Figure 2 polymers-15-02800-f002:**
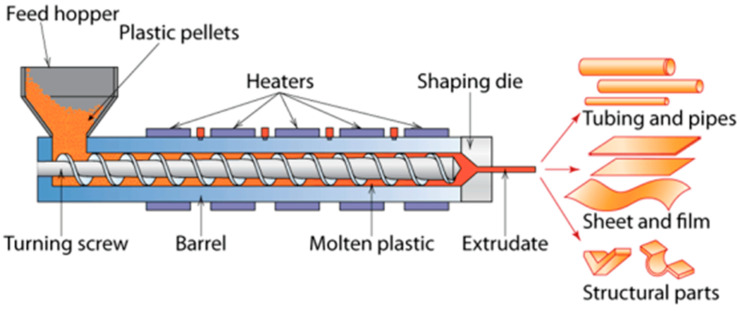
The mechanism of dry casting techniques for edible film formation [9].

## Data Availability

The data will be made available upon request.

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
