# Peer review of "Edible Film Casting Techniques and Materials and Their Utilization for Meat-Based Product Packaging"

_polymers, 2023, doi:10.3390/polym15132800_

Round 1
Reviewer 1 Report
In this manuscript, authors reviewed materials and casting techniques of edible films used for meat packaging. Authors compiled two casting methods and three categories of materials, followed by a summary of advantages and disadvantages.
The content of the manuscript can be a good contribution to the field. But I personally found the language hard to read. The grammars are almost flawless, but the word choices and expressions are ornate and grandiloquent. The language style is pretentious and not for communication purpose in scientific communities. A journal paper is supposed to prioritize clear and concise scientific content, instead of showing off vocabularies. I understand that people may have different writing styles, but please at least tune the style down a bit so that people can enjoy authors’ science other than authors’ vocabularies. Thus, I suggest a major revision mainly focusing on a full language modification along with additional comments below.
Additional Comments:
1. Page 2, Line 56-59: Here, authors introduced the production method for edible films and made it sound like that this is the only method. This is contradictory to Section 2 where different formation methods were summarized. Please revise this sentence to avoid potential confusion.
2. Page 2, Section 2.1: The whole section jumps into polysaccharide film fabrication directly. This section should be about “wet” formation and polysaccharide should only serve as an example. Please add some introduction or description about wet formation in general before talking about polysaccharides.
3. Page 2, Line 93: Please replace “two elements” with “two compounds”.
4. Page 3, Line 110: I am not sure if this is a common expression in the field, but how is a starch film “palatable”? And why is it called “membrane” in this paragraph while the rest of manuscript called it “film”. Please revise the whole manuscript and replace all the “membranes” with “films” if authors meant the same thing.
5. Page 3, Line 135: Please replace “temperature spectrum” with “temperature range”.
6. Page 3, Line 160: Authors mentioned “printing” as one of four steps. There is no “printing” in either Figure 1 or the rest of the paragraph. What does printing mean?
7. Page 4, Line 143: Similar to the comment above, here appears “printing” again with no context. How does an extruder “print” a film?
8. Page 4, Line 156-158: First, there is not a full sentence in grammar. Second, I don’t understand why this conditioning method is described as “innovatively”. This conditioning method takes as long as 7 days comparing with the one above where only two hours are enough.
9. Page 4, Line 161-162: Please specify the time range for each method, so that readers can be convinced by this statement about “time consuming” directly.
10. Page 5, Line 209: The word “co-biopolymer” is not an accurate term. First, author should know the difference between “co-polymers” and “composites”, and then choose a correct word. Second, fats are not polymers.
11. Page 7, Line 278-279: This is not a full sentence. The reference number should not serve as a component in a sentence.
12. Page 7, Line 281: In this manuscript, some species names are italic, and some are not. Please make them all italic for consistency.
13. Page 9, Line 391: Same as above. This is not a full sentence. The reference number should not serve as a component in a sentence. Please check the whole manuscript and make sure to fix all the same problems. There are plenty of them.
14. Page 10, Section 4: The title misses an “and”.
Unnecessarily overcomplex language. Please see main comments and suggestions.
Author Response
Dear Reviewer,
Thank you very much for your positive comments and suggestions. We admit that we still need to simplify the language and we already para-phrase some sentence and follows all of your suggestion. Looking forward to hearing from you.
Best Regards,
Authors

Reviewer 2 Report
1. Section 2.1 : According to my understanding starch films are very brittle. Is it possible to get 100% starch films by just gelatinization or does it require any plasticizer for complete film formation?
2. Line 133: Authors mention it the dry technique requires plasticizers. They should elaborate the types of plasticizers used. Are the starch films still considered edible after using these plasticizers?
Line 204 mentions glycerol? are there any other plasticizers used besides glycerol that are considered edible?
3. Section 3.2 Dry Formation Casting Techniques should labeled as 2.2.
4. Section 3.1.1 Starch. Authors should elaborate on the difference between amylose and amylopectin and how different percentage of those can affect gelatinization temperature.
5. Section 3.1.3 gums should be section 3.1.4
6. As it review talks about different polysaccharides, it might be useful for readers to have chemical/ polymer structures of all all the discussed polysaccharides added as a figure and a description of the monomer units and how they differ from each other.
7. Section 3.2 mentions vegetable oil. This section only contains detailed description of wax under lipids. Addition of different oils as additives to edible films might make this review well rounded. Authors can add details about chain lengths of fatty oils, types, sources, what makes oils rancid not beneficial for packaging application anymore, etc.
The English is well written. The numbering and section need a minor correction as mentioned above.
Author Response
Dear Reviewer,
Thank you very much for your positive comments and suggestions. We admit that we still need to simplify the language and we already para-phrase some sentence and added some explanation to follows all of your suggestion. Looking forward to hearing from you.
Best Regards,
Authors

Round 2
Reviewer 1 Report
Authors considered the previous round suggestions and made revision for all the points I listed. But I am quite disappointed that author did not treat my below comment seriously:
"Same as above. This is not a full sentence. The reference number should not serve as a component in a sentence. Please check the whole manuscript and make sure to fix all the same problems. There are plenty of them."
I pointed out TWO examples of this problem, and explicitly asked a thorough check on the whole manuscript. I don't know if authors are too busy to do it but I can still find at least nine these mistakes by one glance. I value my time and it is not my responsibility to pinpoint every single mistake under one category. I hope authors can respect reviewers' time and comments.
Unnecessarily overcomplex language.
Author Response
Dear Reviewer,
We apologize that we are not carefully response your comment regarding the citation placing on the sentence. We admit that we still need to thoroughly check and recheck the manuscript regarding that point. In occasion with that, we already re-check the manuscript and re-write it. We also done some para-phrase to make sure that the writing style is having a simpler structure and not having over complex language. We hope our correction could accommodate your suggestions.
Thank you very much for your kind review and guidance, looking forward to hearing from you.
Best regards,
Authors